

# Multiple mediation effect of coping styles and self-esteem in the relationship between spousal support and pregnancy stress of married immigrant pregnant women

So-hyun Moon[1] and Miok Kim[2]

[1] Department of Nursing, College of Medicine, Chosun University, Gwangju, Republic of Korea
[2] Department of Nursing, College of Nursing, Dankook University, Cheonan, Chungnam, Republic of Korea

Corresponding author
Miok Kim, aprilsea@dankook.ac.kr

## ABSTRACT

**Background:** The purpose of this study was to identify the total, direct, and indirect influence of spousal support on pregnancy stress among married immigrant pregnant women. The study aimed to determine the relative magnitudes of specific mediating effects of coping styles and self-esteem.

**Method:** A cross-sectional correlational survey was conducted in Jeonnam with 206 married immigrant pregnant women. Data were collected from September 7 to November 7 in 2019. A self-report questionnaire was used to measure spousal support, pregnancy stress, coping styles, and self-esteem. The study employed a linear multiple regression analysis to examine the potential multi-mediating effects. The effect size was set at 0.15, the significance level at 0.05, and the power at 0.95. Through the analysis, the researchers explored the mediating mechanisms among the variables and identified the presence of multi-mediating effects.

**Results:** The effect sizes (b) and statistical significance (p) for the predictors were as follows: problem-focused coping (b = 0.13, $p$ = 0.001), emotion-focused coping (b = 0.11, $p$ = 0.004), and self-esteem (b = 0.10, $p$ < 0.001). Emotion-focused coping (b = 0.26, $p$ = 0.001) and self-esteem (b = −0.20, $p$ = 0.035) had a significant impact on pregnancy stress. The total effect of spousal support on pregnancy stress was significant at −0.25 ($p$ < 0.001), and the direct effect was also significant at −0.26 ($p$ < 0.001). We observed significant mediating effects for emotion-focused coping and self-esteem.

**Conclusions:** As a result of this study, the self-esteem of married immigrant pregnant women can have a protective effect by preventing the aggravation of pregnancy stress in the relationship between spousal support and pregnancy stress. Meanwhile, the emotion-focused coping style can balance out the effect of self-esteem. Therefore, in order to alleviate the stress of pregnancy for women, it is necessary to provide intervention to help improve self-esteem with spousal support. In addition, nursing professionals should help them use appropriate coping styles.

## INTRODUCTION

In 2019, the number of multicultural marriages in South Korea was 24,721, an increase of 4.0% (948 cases) compared to 2018 (*Statistics Korea, 2020*). This accounted for 10.3% of the total number of marriages in South Korea, with the number of marriages between Korean men and foreign women accounting for 69.3% of them (*Statistics Korea, 2020*). Social problems related to migration, such as poor socioeconomic status, discrimination and social exclusion, multiple losses, and chronic stress can have serious negative consequences for the physical and mental health of immigrants (*Hadgkiss & Renzaho, 2014*). Marriage can provide women with economic and social support, but at the same time, it can weaken their control over work and household chores, potentially having both positive and negative effects on their health (*Ballantyne, 1999*). Particularly, married immigrant women who migrated through marriage experience pregnancy and childbirth before they have culturally adapted and experience fears related to unplanned pregnancies and conflicts arising from prenatal cultures different from their native countries (*Kim et al., 2019*). Therefore, it is crucial to provide assistance to married immigrant women in order to facilitate their flexible adaptation to a new culture, prepare them for their roles as stable mothers, and enable them to become healthy and active members of multicultural families (*Kim & Noh, 2018*).

Pregnancy is a time of significant change for many women (*Corbijn van Willenswaard et al., 2017*) and, according to a large-scale US study, about 84% of women experience some level of stress during pregnancy, and 6% experience a high level of stress (*Woods et al., 2010*). Pregnancy stress can influence fetal malformations and the pregnant woman's comfort, and it can directly affect fetal development and interfere with normal development (*Dipietro, 2012*). Thus, not only does stress during pregnancy increase negative birth outcomes such as low birth weight, premature birth, and unplanned cesarean delivery (*Coussons-Read et al., 2012*), but prenatal stress has a wide range of negative effects during infancy, childhood, and adulthood through neuroendocrine, immune, cardiovascular, metabolic, and behavioral pathways (*Lobel & Dunkel Schetter, 2016*). Specifically, for women of poor economic status and minority races, the prenatal period is affected by numerous physical, emotional, social, and financial stresses (*Bloom et al., 2013*) which can negatively affect perinatal health outcomes (*Kita et al., 2015*). In order to take such various factors into account, a multi-faceted factor analysis for the well-being of married immigrant women during pregnancy is needed.

Interpersonal factors can promote emotional disorders in women who experience emotional difficulties during pregnancy (*Diaz et al., 2007*). Here, the support of affectionate spouses in establishing and adapting to a new family in South Korea is the most important protective factor for pregnant women's emotional experiences (*Rini et al., 2006*). The level of support from a spouse plays a crucial role in reducing the stress levels perceived by women during pregnancy (*Bedaso et al., 2021*). If the spouse or family support system is lacking or unsatisfactory, married immigrant women become vulnerable

to stress during pregnancy (*Jesse et al., 2009*). These studies support the concept that the dynamics of marriage migration and subsequent spousal relationships are closely associated with women's experiences of pregnancy stress. Therefore, it is necessary to maintain and promote the mental well-being of pregnant women through appropriate support during pregnancy.

Self-esteem is a psychological resource that can buffer negative experiences in life, including those related to physical and mental well-being (*Cast & Burke, 2002*). It can also mediate the relationship between cultural conflict and negative emotions (*Verkuyten & Nekuee, 1999*). Given that low self-esteem in immigrants is associated with high levels of depression and poor mental health (*Zeiders, Umaña-Taylor & Derlan, 2013*), it is crucial for married immigrant women to maintain positive self-esteem during pregnancy in order to minimize mental health problems such as stress.

Coping mechanisms are strategies that help individuals deal with stress, and coping styles play an important role in their overall well-being (*Aldwin & Revenson, 1987*). In other words, since levels of stress can vary depending on the coping styles used by individuals (*Gourounti, Anagnostopoulos & Lykeridou, 2013*), the coping styles used by individuals in stressful situations like marriage migration can control the level of additional stress that occurs during pregnancy. When individuals use the problem-focused coping style, their stress levels decrease and their mental health improves (*Charsouei et al., 2021*), whereas the emotion-focused coping style is associated with higher levels of negative emotions during pregnancy (*Aghayousefi et al., 2012*) and negative childbirth prognoses including postpartum depression and premature birth (*Falah-Hassani et al., 2015*). Here, the emotion-focused coping style is the most commonly used coping method by pregnant women who complained of psychological difficulties such as fear and stress during the COVID-19 pandemic (*Rimal, Thapa & Shrestha, 2022*).

Pregnancy stress is more effective in predicting pregnancy outcomes than latent factors representing general and life event stress perceived by individuals (*Lobel et al., 2008*). Since pregnancy stress is variously influenced, identifying factors contributing to pregnancy stress can play an important role in the mental health of pregnant women (*Dolatian et al., 2013*). Here, most effects or phenomena work simultaneously through multiple mechanisms; thus, no model can completely and accurately explain the outcome variable. If there is a reasonable reason that antecedent variables affect outcome variables through multiple mechanisms, estimating a model that allows multiple processes to occur simultaneously may be a more effective method (*Hayes, 2015*). Based on a previous study (*Razurel et al., 2013*) which highlighted the importance of social support and positive coping for pregnant women in alleviating psychological distress, our research aims to examine the significant multi-mediating effects of stress coping style and self-esteem among married immigrant women. Specifically, we focus on understanding how spousal support influences pregnancy stress. We consider pregnancy stress as an outcome variable influenced by individual coping styles within the broader context of stress related to migration.

# MATERIALS AND METHODS

## Research design

This study is a cross sectional study to determine the mediating effects of coping and self-esteem in the relationship between spouse support and pregnancy stress perceived by marriage migrant pregnant women.

## Subjects

Convenience sampling was conducted for pregnant women who received prenatal care at Miz-I Hospital and Hanlove Hospital, two women's hospitals located in Jeonnam. The selection criteria included women who understood the purpose of the study and agreed to participate; had moved to South Korea for marriage and were currently pregnant; had lived in South Korea for less than 6 years; and were able to communicate in Korean or understood Vietnamese, Chinese, Filipino, or English. The exclusion criteria involved individuals undergoing treatment for physical or mental health conditions and receiving hospitalization for pregnancy-related complications.

The number of samples was calculated using the G Power 3.1.9.4 program. Considering a previous study (*Kang & Han, 2016*) with pregnancy stress as the dependent variable, an effect size of 0.15, a significance level of 0.05, and a power of 0.95, a total of 129 participants were calculated to be the required sample size for the multiple linear regression analysis, taking into account the predictor variables of spousal support, self-esteem, active coping, and passive coping. A questionnaire was distributed to 230 people, considering the minimum sample size, response rate of participants from multiple countries, and dropout rate. Of the 230 copies retrieved, 206 (dropout rate: 6.1%) were used for the final analysis, excluding 24 with some skipped or double-checked for a single question.

## Measurements

The questionnaire was translated into five languages for subjects whose mother tongues were Korean, Vietnamese, Chinese, Filipino, and English. For the primary translation, a total of four people, one from the multicultural family support center who was fluent in Korean and the language of each country, performed the primary translation. Afterward, two native speakers of each language who had lived in South Korea for more than 10 years and were fluent in Korean performed reverse translation at the center to ensure the accuracy of the translation. For the translated questionnaire, a preliminary survey was conducted on five married immigrant women for each language, and the suitability of the questionnaire was reviewed to ensure that the subjects sufficiently understood it and were able to complete it. As a result of conducting a preliminary survey of those who are fluent in Korean and the language of the country, and examining whether there are any differences in expressions and vocabulary choices, it was decided to use all the items as they are because there was no change in meaning. In this study, the reliability of the tool Cronbach's α value was similar to that of the original tool. However, it is necessary to

ensure the objective reliability and validity of the various questionnaires translated in the future.

General characteristics include information on age, spouse's age, nationality, last formal education, job, spouse's job, living with spouse, monthly family income, currently having children, total number of pregnancies, current gestational age, complications of current pregnancy, time spent conversation with spouse per day.

### Spousal support

For spousal support, 11 items related to spousal support were used from 22 items (11 items on family and friend support were excluded) from the social support scale of the Prenatal Psychosocial Profile (PPP) developed by *Curry, Campbell & Christian (1994)* for pregnant women, translated by *Kim (2000)* after correcting and supplementing through a preliminary survey. The content and composition validity were verified by experts. The items were scored on a 6-point Likert scale (1 = "very dissatisfied" to 6 = "very satisfied") with a higher score indicating a higher level of spousal support. In *Curry, Campbell & Christian*'s *(1994)* study, Cronbach's alpha, indicating the reliability of social support, was 0.97.; in *Kim*'s *(2000)* study, that of spousal support was 0.96.; and in this study, it was 0.96.

### Pregnancy stress

Pregnancy stress was evaluated by using 11 items from the PPP (*Curry, Campbell & Christian, 1994*; *Kim, 2000*), scored on a 4-point Likert scale (1 = "not stressful at all" to 4 = "very stressful") with a higher score indicating a higher level of stress. Cronbach's alpha was 0.70 in *Curry, Campbell & Christian*'s *(1994)* study and 0.92 in this study.

### Self-esteem

Self-esteem was evaluated by using 11 items on self-esteem and acceptance from the PPP (*Curry, Campbell & Christian, 1994*; *Kim, 2000*). The items were scored on a 4-point Likert scale (1 = "not at all" to 4 = "always") with a higher score indicating a higher level of self-esteem during pregnancy. Cronbach's alpha was 0.80 for the self-esteem scale of the PPP by *Curry, Campbell & Christian (1994)*, and 0.75 in this study.

### Coping styles

Coping styles were measured by using 18 items of a scale developed by *Billings & Moos (1981)*, translated and modified by *Kim (1989)*, and modified by *Kim (2003)* to suit pregnant women's situations. Eight items on the problem-focused coping style and 10 items on the emotion-focused coping style were scored on a 4-point Likert scale (1 = "strongly disagree" to 4 = "strongly agree") with a higher score indicating a higher frequency of using the coping styles. Cronbach's alpha, indicating reliability, was 0.62 in *Billings & Moos*'s *(1981)* study, 0.75 in *Kim*'s *(1989)* study, and 0.81 in *Kim*'s *(2003)* study. In this study, Cronbach's alpha was 0.79 for the problem-focused coping style and 0.74 for the emotion-focused coping style.
## Data collection

Data collection was conducted between September 7 and November 7, 2019, after obtaining written consent from participants who expressed their intention to participate in the study after the researcher and research assistant explained the purpose and method of the study with the approval of the institution. Questionnaires were collected face-to-face in article form.

If the questionnaire was not translated into the participants' native language, the Korean questionnaire was administered. The distribution took place when the scores in all areas reached or exceeded the "average" level of three points. The participants' level of Korean communication ability (*Lee et al., 2013*) was evaluated using a 5-point scale (1 = "not at all" to 4 = "very good"), encompassing four items related to speaking, listening, reading, and writing. The completion time for the questionnaire was approximately 15 to 25 min.

All participants were informed that they could pull out at any time during the study and that the collected data would not be used for any purpose other than research and would be processed into codes and used as computer data. Participants who completed the survey were given a compensation of wet wipes and bottled water valued at $5 as a token of appreciation for their dedicated time and participation in the study. The details of the research incentive were not revealed before the participants completed the questionnaire, ensuring that their decision to participate was purely voluntary and self-determined. This study was approved by the Institutional Review Board (IRB) of Chosun University (IRB-2-1041055-AB-N-01-2019-11).

## Data analysis

The collected data were analyzed using SPSS version 25.0 (SPSS, inc., Chicago, IL, USA) and indirect SPSS macros for multiple mediations (*Preacher & Hayes, 2008*). Descriptive statistics were used for general characteristics and frequency, percentage, mean, and standard deviation of measurement variables, and Pearson's correlation coefficient was used for the correlation between variables. To analyze the multiparameter model, which is the hypothetical model of this study, a bootstrapping procedure was conducted using an indirect SPSS macro. The size and significance of indirect effects were analyzed using the 95% Confidence Interval (95% CI). If the 95% confidence interval for the estimate of the indirect effect does not contain zero, the indirect effect was statistically significant at the 0.05 level.

# RESULTS

## General characteristics of the subjects

The study participants had an average age of 27.12 years (SD 5.80), while their spouses averaged 41.32 years (SD 7.98). The majority were from Vietnam (77.2%) and the Philippines (14.1%) before marriage migration. Education level was mostly middle school or high school (77.7%), and unemployment rate was 82.0%. Participants lived with their spouses, who were primarily office workers (39.3%) or self-employed (36.9%) (Table 1).

**Table 1 General characteristics of subjects (N = 206).** Each variable was described as frequency, percentage, mean and standard deviation. 'Won' is a unit of Korean money.

| Characteristics | Categories | N (%) or Mean ± SD |
|---|---|---|
| Age (year) | | 27.12 ± 5.80 |
| Spouse's age (year) | | 41.32 ± 7.98 |
| Nationality | Mongolia | 2 (1.0) |
| | Vietnam | 159 (77.2) |
| | Japan | 3 (1.5) |
| | China | 10 (4.9) |
| | Cambodia | 3 (1.5) |
| | Philippines | 29 (14.1) |
| Last formal education | ≤Elementary school | 17 (8.3) |
| | Middle school & high school | 160 (77.7) |
| | ≥College | 29 (14.1) |
| Job | None | 169 (82.0) |
| | Have | 37 (18.0) |
| Spouse's job | Office worker | 79 (39.3) |
| | Self-employed | 76 (36.9) |
| | other | 51 (24.8) |
| Living with spouse | Yes | 204 (99.0) |
| | No | 2 (1.0) |
| Monthly family income (Korean won) | <1,000,000 | 6 (2.9) |
| | ≥1,000,000–<2,000,000 | 59 (28.7) |
| | ≥2,000,000 | 80 (38.8) |
| | Don't know | 61 (29.6) |
| Currently having children | Have | 94 (45.6) |
| | Do not have | 112 (54.4) |
| Total number of pregnancies | | 2.01 ± 1.11 |
| Current gestational age | | 30.32 ± 5.31 |
| Current pregnancy | Planned | 187 (90.8) |
| | Unplanned | 19 (9.2) |
| Complications of current pregnancy | Yes | 11 (5.3) |
| | No | 195 (94.7) |
| Time spent conversation with spouse per day (hour) | No conversation | 5 (2.4) |
| | <1 | 79 (38.4) |
| | ≥1–<2 | 39 (18.9) |
| | ≥2 | 83 (40.3) |

## Levels of spousal support, pregnancy stress, coping styles, and self-esteem

Spousal support was 4.92 on a 6-point Likert scale and self-esteem was 3.19 on a 4-point scale, both above moderate. Pregnancy stress was slightly below moderate at 1.59 on a 4-point Likert scale. In terms of coping styles, the emotional-focused style had a slightly higher mean score than the problem-focused style (Table 2).

**Table 2 Levels of spousal support, pregnancy stress, self-esteem, and coping styles ($N = 206$).** The range of scale and mean and standard deviation for each variable.

| Variables | | Range of scale | Mean ± SD |
|---|---|---|---|
| Spousal support | | 1~6 | 4.92 ± 1.16 |
| Pregnancy stress | | 1~4 | 1.59 ± 0.81 |
| Self-esteem | | 1~4 | 3.19 ± 0.42 |
| Coping style | Problem-focused | 1~4 | 2.48 ± 0.65 |
| | Emotional-focused | 1~4 | 2.79 ± 0.64 |

**Table 3 Correlation between spousal support, pregnancy stress, self-esteem, and coping styles ($N = 206$).** The correlation between each variable was written with three decimal point.

| Variables | | Spousal support | Pregnancy stress | Self-esteem | Coping style | |
|---|---|---|---|---|---|---|
| | | | | | Problem-focused | Emotional-focused |
| | | r (p) | r (p) | r (p) | r (p) | r (p) |
| Spousal support | | 1 | | | | |
| Pregnancy stress | | −0.479 (<0.001) | 1 | | | |
| Self-esteem | | 0.263 (<0.001) | −0.180 (0.009) | 1 | | |
| Coping style | Problem-focused | 0.231 (0.001) | 0.016 (0.813) | 0.312 (<0.001) | 1 | |
| | Emotional-focused | 0.203 (0.003) | 0.114 (0.100) | 0.335 (<0.001) | 0.702 (<0.001) | 1 |

## Correlation of spousal support, pregnancy stress, coping styles, and self-esteem

Table 3 presents the results of the bivariate correlation analysis of the association between spousal support, self-esteem, coping styles, and pregnancy stress. Pregnancy stress showed a statistically significant inverse association with spousal support and self-esteem. Specifically, the self-esteem and pregnancy stress correlation was smaller (−0.180) than the correlation between spousal support and pregnancy stress(−0.479). Pregnancy stress did not show a statistically significant correlation with problem-focused coping style or emotion-focused coping style. Spousal support was found to have a positive correlation with problem-focused coping style, emotion-focused coping style, and self-esteem (Table 3).

## Multiple-mediation estimates for pregnancy stress

We found statistically significant effects of spousal support on various parameters. The effect sizes (b) and statistical significance (p) are as follows: problem-focused coping (b = 0.13, p = 0.001), emotion-focused coping (b = 0.11, p = 0.004), and self-esteem (b = 0.10, p < 0.001). Specifically, emotion-focused coping (b = 0.26, p = 0.001) and self-esteem (b = −0.20, p = 0.035) were found to have a significant impact on pregnancy

stress. The total effect of spousal support on pregnancy stress was significant at −0.25 ($p < 0.001$), and the direct effect was also significant at −0.26 ($p < 0.001$). While the overall indirect effect, which represents the combined effects of different parameters, was not statistically significant, there were significant mediating effects observed for emotion-focused coping and self-esteem. Excluding the first parameter, problem-focused coping, the mediating effects were 0.03 (95% CI [0.01–0.08]) and −0.02 (95% CI [−0.05 to −0.01]) for emotion-focused coping and self-esteem, respectively. Both emotion-focused coping and self-esteem showed statistically significant effects, indicated by the 95% confidence intervals that did not contain zero and had consistent directions. This suggests that emotion-focused coping and self-esteem mediate the relationship between spousal support and pregnancy stress. Additionally, we observed a pure mediating effect of 0.01 which represents the difference between the two mediating effects (emotion-focused coping of 0.03 and self-esteem of −0.02). Notably, emotion-focused coping had a positive effect, while self-esteem had a negative effect, canceling each other's mediating effects (Table 4 and Fig. 1).

## DISCUSSION

We found that the total effect of the degree of spousal support perceived by married immigrant women on pregnancy stress was significant, confirming the results of a previous study (*Seo et al., 2020*) that reported a high awareness of spousal support during pregnancy contributed to the reduction of pregnancy stress. As the negative emotional experiences of pregnant women make it difficult for them to have positive attitudes toward pregnancy and childbirth, perceiving it as a relative burden, these experiences can act as obstacles to establishing an appropriate maternal identity (*Seo et al., 2020*). Here, spousal support is an important factor for health and adaptation during pregnancy (*Hwang, 2019*) and it plays a role in controlling or buffering stress that affects the physical and psychological health of pregnant women (*Yilmaz et al., 2021*). Therefore, regarding marriage migration, under the support system of the community, education for spouses is required to help them establish themselves as appropriate supporters based on a sufficient understanding of the experience of immigrant women's adaptation process, plan pregnancy together, and provide appropriate help during pregnancy.

Meanwhile, despite the recommendation of the American College of Obstetricians and Gynecologists (*American College of Gynecologists, 2006*) that the psychosocial assessment of pregnant women should be included as part of routine prenatal care, it is not widely practiced (*Kingston et al., 2012*), and immigrant women are also reluctant to seek professional help for mental health problems (*Guzder, Santhanam-Martin & Rousseau, 2014*). Therefore, hospital support programs and related organizations in the community are required to select and support immigrant women who are vulnerable to mental health problems due to psychosocial risk factors related to migration.

In this study, the multi-mediating effect of individual coping styles and self-esteem was examined in the relationship between spousal support and pregnancy stress perceived by married immigrant women. As a result, the problem-focused coping style—active cognitive and behavioral efforts to solve a certain problem—had no mediating effect,

**Table 4 Multiple-mediation estimates for pregnancy stress ($N = 206$).**

| Variables | Coeff | se | t | p | LLCI | ULCI |
|---|---|---|---|---|---|---|
| Effects of spousal support on mediators | | | | | | |
| Problem-focused coping | 0.13 | 0.04 | 3.37 | 0.001 | 0.05 | 0.20 |
| Emotion-focused coping | 0.11 | 0.03 | 2.91 | 0.004 | 0.04 | 0.18 |
| Self-esteem | 0.10 | 0.02 | 3.91 | <0.001 | 0.05 | 0.14 |
| Effects of mediators on pregnancy stress | | | | | | |
| Problem-focused coping | −0.02 | 0.07 | 0.22 | 0.824 | −0.17 | 0.13 |
| Emotion-focused coping | 0.26 | 0.08 | 3.22 | 0.001 | 0.10 | 0.42 |
| Self-esteem | −0.20 | 0.09 | −2.11 | 0.035 | −0.32 | −0.19 |
| | **Effect** | **se** | **t** | **p** | **LLCI** | **ULCI** |
| Total effect of spousal support on pregnancy stress | −0.25 | 0.03 | −7.68 | <0.001 | −0.31 | −0.18 |
| Direct effect of spousal support on pregnancy stress | −0.26 | 0.03 | −7.78 | <0.001 | −0.32 | −0.19 |
| Indirect effect of spousal support on pregnancy stress (total) | 0.01 | 0.01 | | | −0.02 | 0.04 |
| Problem-focused coping | −0.00 | 0.01 | | | −0.02 | 0.02 |
| Emotion-focused coping | 0.03 | 0.02 | | | 0.01 | 0.08 |
| Self-esteem | −0.02 | 0.01 | | | −0.05 | −0.01 |

Notes:
CI, confidence interval; LL, lower limit; UL, upper limit; bootstrap samples = 5.000.
$R^2 = 22.54$, F = 59.06, $p < 0.001$. The results are described in the table.

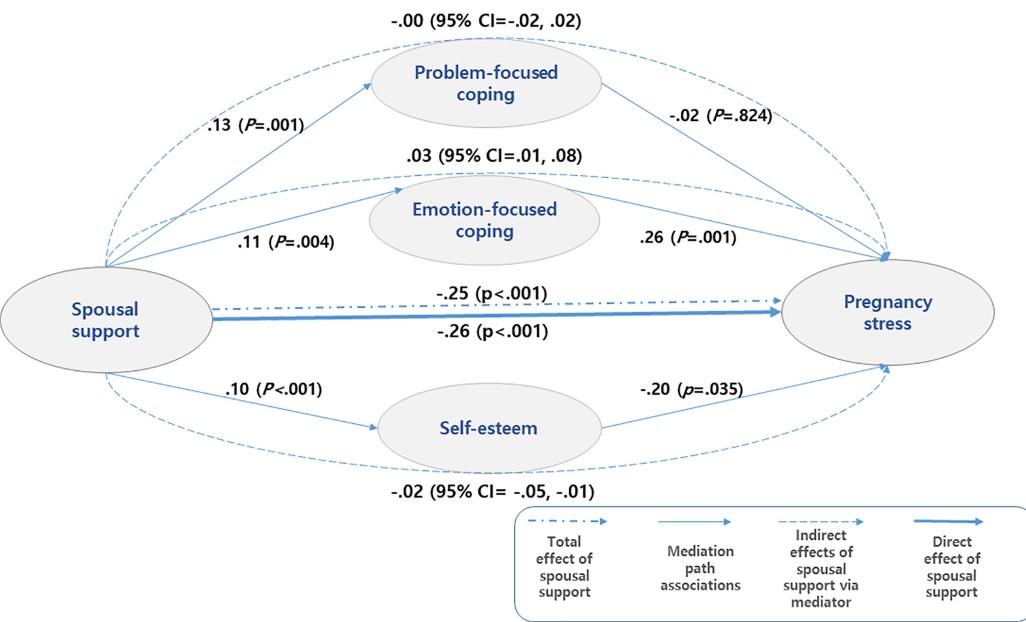

**Figure 1 Multiple-mediation bootstrap analysis of relationships between spousal support and pregnancy stress as mediated by problem-focused coping, emotion-focused coping, and self-esteem in the total sample.**

whereas the emotion-focused coping style and self-esteem—which regulated emotional responses to problems—had a mediating effect. This suggested that while married immigrant women used both problem-focused and emotion-focused coping styles upon

perceiving spousal support as high, using the emotion-focused coping style more at this time could aggravate pregnancy stress. This contradicted results from a previous study (Sarani et al., 2015) where problem-focused coping was appropriate as a stress relief strategy during pregnancy but was in line with the report (Baral & Bhagawati, 2019) that passive coping, such as the emotion-focused coping style, was associated with psychological pain. As the physical and psychological changes caused by pregnancy cause high levels of stress in women, it is very important to use appropriate coping strategies (Sarani et al., 2015). Nurses should help women positively control stress that may occur during pregnancy by minimizing negative coping mechanisms in which subjects avoid the problem and encouraging active problem solving and positive emotional coping styles such as active participation in pregnancy and childbirth-related education, active communication with family members, and self-help group participation.

On the other hand, each of problem-centered and emotion-centered coping styles can be interpreted in a bipolar dimension. For problem-centered coping styles, a high level of problem coping can indicate active problem solving and a low level of problem coping can indicate problem avoidance. For emotion-centered coping styles, individuals can regulate their emotional responses to problems through positive or negative emotional coping (Gol & Cook, 2004). Thus, positive and negative coping are not mutually exclusive and can be used simultaneously, and rather than using one specific coping behavior, integrating various coping behaviors can promote women's abilities to relieve pregnancy-related stress (Yu et al., 2020). Since this study did not utilize a tool to represent problem-focused coping style and emotion-focused coping style as opposite ends of a single dimension, which would have allowed for analyzing the mediating effects on each coping style's continuum, we cannot exclude the possibility that the two forms of problem-focused coping and emotion-focused coping produced different outcomes in our results. However, it is important to recognize that individuals' coping styles can have contrasting aspects that are effective in alleviating stress during pregnancy. Therefore, it is necessary to assess women's coping styles beforehand and assist them in establishing appropriate coping strategies to contribute to stress relief.

The multi-mediating effect analysis in this study showed that self-esteem mediated the relationship between spousal support and pregnancy stress, and spousal support during pregnancy had a positive effect on pregnancy stress through the self-esteem of the pregnant woman. While immigrant women experience many changes in the process of settling in South Korea through marriage, pregnancy can both increase women's stress (Gol & Cook, 2004) and increase self-esteem (Yesilcinar, Yanik & Akbulut, 2020). Women's self-esteem is enhanced by the experience of emotional support and care from their spouse, which helps them cope effectively with stressful situations (Taylor et al., 2008) and helps them overcome negative life events (Lee & Lee, 2012). On the other hand, negative expectations for the future, unplanned pregnancy, and low self-esteem can lead to feelings of hopelessness and decreased self-confidence (Fedorowicz et al., 2014). Therefore, married immigrant pregnant women should recognize that improving their self-esteem can prevent the increase of stress in various changes related to pregnancy. Here, nurses should help immigrant women positively perceive the changed circumstances related to

pregnancy and actively participate in prenatal management, such as prenatal education and practical prenatal care education programs.

In this study, since the direct effect of the degree of spousal support perceived by married immigrant women on pregnancy stress was quite large, the multi-mediating effect of individual coping styles and self-esteem was not as large. Nevertheless, since the mediating effects of the problem-focused coping style, the emotion-focused coping style, and self-esteem were not the same and the directions of their effects were different, we can suggest the direction of nursing interventions to help married immigrant women during pregnancy. Here, although the direct effect of the relationship between spousal support and pregnancy stress was large and limited in this study, a nursing strategy that promoted the self-esteem of pregnant women, which showed the same direction as the direct effect of spousal support, could be an important factor for maximizing the effect of nursing interventions to relieve pregnancy stress. For married immigrant women, spousal support helps them recover self-esteem that has deteriorated due to the stress from acculturation (*Kim, Lim & Jeong, 2013*). Nurses should seek out measures to encourage the individual efforts of married immigrant women, as well as spousal support strategies, to enhance their self-esteem. Additionally, although the emotion-focused coping style has a mediating effect, the direction of its influence is different from that of self-esteem, thereby offsetting the effect of self-esteem in the relationship between spousal support and pregnancy stress. In other words, recognizing that negative emotion-focused coping, such as avoiding problems, denying reality, and running away from reality in various changing situations, can increase pregnancy stress despite a high perceived level of spousal support, efforts increase the use of positive and effective coping styles.

To prevent aggravation of pregnancy stress in married immigrant women, individuals, families, and members of society should have a high awareness of the need for spousal support. Since a pregnant woman's high level of self-esteem can contribute to relieving stress during pregnancy, spousal support strategies that can improve the self-esteem of the pregnant woman should be sought. In this study, there could be a need for closer control of the number of pregnancies and births, current gestational weeks, and length of stay in South Korea, and our data was limited as we studied married immigrant pregnant women residing in one region. Therefore, generalization of the research findings may be limited, and the possibility of measurement errors arising from subjective interpretation and recall bias of the survey respondents cannot be excluded. Moreover, it is important to note that the perceived level of spousal support among married immigrant pregnant women was found to be very high, while the level of pregnancy stress was measured at a low level. This suggests the potential influence of unconsidered latent factors that could impact the results. Nevertheless, this study is meaningful in that it provides an understanding of pregnancy stress experienced by married immigrant pregnant women and a direction for nursing interventions by identifying the relationship between spousal support and pregnancy stress and revealing the multiple mediating effects of self-esteem and coping style.

## CONCLUSIONS

This study is a correlational study to confirm the multi-mediating effect of stress coping style used by pregnant women and their self-esteem in the relationship between spousal support and pregnancy stress in married immigrant women. Spousal support had a significant total, direct and indirect effect on pregnancy stress. The multi-mediating effect of emotion-focused coping style and self-esteem was confirmed on the relationship between spousal support and pregnancy stress.

To prevent the aggravation of pregnancy stress in marriage migrant women, active support from spouses is crucial. In order to facilitate this, healthcare professionals should strive to raise awareness among spouses regarding the importance of providing support during pregnancy. In particular, the high self-esteem of the pregnant woman can contribute to relieving stress during pregnancy. Therefore, while providing interventions to help improve self-esteem in pregnant women, it is necessary to explore spousal support strategies that can improve self-esteem in pregnant women. In the future, the results of this study are expected to be used as basic data for nursing interventions to help married immigrant women plan pregnancies with their spouses and minimize pregnancy stress through appropriate support from their spouses, families, and communities during pregnancy. Considering that the degree of spousal support perceived by married immigrant pregnant women was higher than average in the results of this study, it is suggested to evaluate the level of social support and pregnancy stress, including family support. It is also suggested to analyze the multimediating effect of coping styles using a tool that can be divided into the bipolar dimensions of problem-focused and emotion-focused coping styles.

There is a growing interest in married immigrant women as relatively vulnerable subjects, such as getting pregnant before adapting to Korean culture. so this study was conducted only on married migrant pregnant women. However, in a future study, it is suggested to compare the characteristics of Korean women and married migrant women.

### Funding
This study was supported by a research fund from Chosun University (2021). The funders had no role in study design, data collection and analysis, decision to publish, or preparation of the manuscript.

### Grant Disclosures
The following grant information was disclosed by the authors:
Chosun University: 2021.

### Competing Interests
The authors declare that they have no competing interests.

## Author Contributions

- So-hyun Moon conceived and designed the experiments, performed the experiments, analyzed the data, prepared figures and/or tables, authored or reviewed drafts of the article, and approved the final draft.
- Miok Kim performed the experiments, analyzed the data, prepared figures and/or tables, authored or reviewed drafts of the article, and approved the final draft.

## Ethics

The following information was supplied relating to ethical approvals (*i.e.*, approving body and any reference numbers):

This study was approved by the Institutional Review Board (IRB) of Chosun University (IRB-2-1041055-AB-N-01-2019-11). Informed consent was obtained from the participants.

## Data Availability

The raw data is available in the Supplemental Files.

## Supplemental Information

Supplemental information for this article can be found online at http://dx.doi.org/10.7717/peerj.16295#supplemental-information.

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
