# Peer review of "Multiple mediation effect of coping styles and self-esteem in the relationship between spousal support and pregnancy stress of married immigrant pregnant women"

_PeerJ, doi:10.7717/peerj.16295_

## Round 0.1 · original submission · Major Revisions

Thank you for submitting your manuscript to the PeerJ. We very much appreciate the time and effort that has gone into the preparation of this article. However, we regret to say that in its current form it is not considered suitable for publication in the PeerJ. The reasons for this decision can be found in the enclosed reports. In general you will see that the reviewers felt positively about the subject matter, however two reviewers found significant omissions detailed below. If however you feel that you could address all of the points raised then we would be willing to consider a revised manuscript. If you choose this route you should provide a full and detailed rebuttal to all points raised.

Reviewer 1 ·

Basic reporting

The language needs some refinement. I share some examples of portions I did not understand well. E.g., The first sentence of the main body of the manuscript needs to mention the main area under study like stress in pregnancy; "A multilateral review is needed to control pregnancy stress in married immigrant women during pregnancy" (line 58-59). It is not clear what multilateral means and how this connects with the contents of the document; the authors often start sentences with "Here,..." but it is not clear what they refer to when they do this; some terms used are a bit obsolete, like "mental diseases" (Lines 107-108); there are some reporting errors - E.g., Lines 113-115: The difference between 230 and 206 is 24, not 14. The reported numbers do not add up; some sentences are very complex and difficult to understand. They need to be rewritten as 2-3 simple sentences. E.g. Lines 89-94, lines 158-162, lines 212-215 (some discrepancy with lines 203-210); some terminologies like "bipolar" (lines 270-280) are confusing; some portions are very prescriptive and not optimal for research communication - lines 70-71, lines 293-294; some grammar issues - “is important” and not “have an important” in lines 72-73; lines 102-103 – M Hospital and H hospital – that makes it two hospitals. Then how is this “a women’s hospital”?; abbreviations need to be expanded (for non-Korean readers KRW may not make sense, and the US dollar equivalent can be given) - line 189;
Coverage of literature review is good, and the background or context is also described reasonably well. The tables are okay, but there is a lot of repetition of contents of the table in the text. The figure shared to reviewers has some text in small font sizes that is difficult to read. The results are generally in line with the hypotheses stated but with some loose generalizations at places.

Experimental design

This part has some issues that need rectification before the manuscript can be considered for further review/ publication. I suggest the authors refer to some standard guidelines for reporting research (STROBE, EQUATOR) as they may help organize the text systematically.
The conceptualization is okay but the authors need to consider the gendered nature of spousal relationships in more detail. I suggest reading Ballantyne (Ballantyne PJ. The social determinants of health: a contribution to the analysis of gender differences in health and illness. Scandinavian Journal of Public Health. 1999 Oct;27(4):290-5.) and rewriting this section, maybe using more geographically or temporally closer references.
The methods section needs clarity in the sample size estimation part and the analysis part. The meaning and the basis of the effect size used in the sample size computation needs to be mentioned. How many persons had to be approached to get a sample size of 230, and what are the features of those who got excluded from the 230 sample size? The model building process needs more clarity – outcome variable, predictors included, adjustments, if any, analytical approaches. This can be mentioned briefly in the abstract also rather than the software used. Briefly mention the basis of the analysis, how mediation was looked for and how the analysis led to the conclusion on multi-mediating effects.
Lines 166-167: Would this gift have coerced participants to give consent? A mention is needed on the nature of the gift. It is better to mention this as a token of gratitude or appreciation, but the authors have to make it clear whether it would have enticed women with less resources to participate.

Validity of the findings

Table 2 – How were the scale ranges (1-6) or (1-4) arrived at as all scales had multiple items according to the methodology section? What was the basis for arriving at this range and why these were used for further analysis?
Lines 325-328 – There is a vague mention of the limitations. This is not enough. The possibility of selection bias, measurement error, limitations of analysis and the impact they are likely to have on the conclusions have to be mentioned.
Lines 340-342 – There are limited results to conclude need for high awareness of spouse, family or community supported as these were not measured or reported on. It is best to keep conclusions within the study findings, although the discussion sections may include reflections beyond the findings
Data have been provided for reference.
The topic is relevant and this work can be used to inform further research or practice but replicability is difficult with the information currently given in the manuscript.

Additional comments

The authors have selected a very relevant topic for research. The initiative is very good and they have put a lot of effort into the topic. They have also made considerable effort to bring clarity to the measurement part of their efforts but the participant selection part and the analysis and results need considerable rewriting. I sincerely hope that the authors put the extra effort to develop this manuscript into a publishable one and share their findings for the benefit of other researchers in this and related discipline.

Reviewer 2 ·

Basic reporting

The manuscript is excellent written and easy to follow and understand. Two simple mistakes were identified:
- line 122: A period is missing at the end of the sentence.
- line 242: There is a wrong hyphen in the word “supporters”.

Experimental design

The study design and underlying research question are well described. Some additional information may improve the comprehensibility and reproducibility:

1. You reported that 14 responses were excluded due to some skipped or duplicates responses (line 11-115). What does duplicate responses mean in this context? Did the women response twice? How was this possible? Maybe you can add a one or two sentences on how the survey was conducted (paper-based or online) and how it was distributed to the women (at what point of their hospital visit or was the questionnaire sent home?).

2. It is excellent that you provided the questionnaire in different languages in the supplementary material. To make the reader aware of this information, please consider referring to the supplementary material on suitable positions in the manuscript.

3. The questionnaire in the supplementary material includes next to the scales described in section 3 also general characteristics. Please also add a short section on this information in the method section.

4. The participants received an incentive for participating (pre-selected gift). Can you add a value range of the gifts or some examples to get an impression of their value?

Validity of the findings

no comment

Additional comments

The discussion section would benefit from a critical appraisal of the methods used and its influence on the generalization of the results.
It is praiseworthy that you provided the questionnaire in several languages. Nevertheless, it cannot be concluded from the paper, whether the population included in this study is representative according to the origin country of the migrant women for South Korea or if there may be a selection bias by systematically excluding a relevant migration population due to the languages used. It might be cleared with comparing the participants characteristics with characteristics of migrant pregnant women, generally in South Korea.

Reviewer 3 ·

Basic reporting

Dear Authors;

Thank you for your interesting study. I have reviewed your manuscript in detail.
Here, there are some comments to improve your manuscript.

Abstract:
This section is written too poorly. The "methods" should be written in more detail. Additionally, the "results" is not appropriate. Please state the values in detail.

Introduction:
The literature review should be written in more detail.

Experimental design

Methods:

1- State the study type.
2- Did you collect the data in an online form or in a paper form?
3- How did you evaluate the reliability and validity of different translated questionnaires?

Validity of the findings

Good

Additional comments

Please state the study limitations.

---

## Round 0.2 · Minor Revisions

Thank you for the update. However, there are still concerns that prevent me from accepting the revised paper. Please pay attention to the reviewers' comments and respond to them carefully.

Reviewer 1 ·

Basic reporting

The article reads considerably better after the revisions.
Some of my concerns still need to be addressed.

1) Lines 65-66: "A multilateral review is needed to control pregnancy stress in married immigrant women during pregnancy."
What does this mean - involvement of multilateral stakeholders? multilateral institutions? multilateral sources of information?
Also, how is this relevant to the rest of the manuscript?

2) Much of the contents of the tables are repeated as such in the text - Sections 2 and 3 of results and tables 2 and 3. At least for the portions in the text that are repetitions of Table 2, perhaps some extra information can be attempted. E.g. do these values indicate high or low levels of the studied variables?

3) Problem-focused and emotion-focused coping styles may overlap considerably and are not necessarily bipolar.

Experimental design

No comment

Validity of the findings

No comment

Additional comments

No comment.

Reviewer 3 ·

Basic reporting

N.A

Experimental design

N.A

Validity of the findings

N.A

Additional comments

N.A

---

## Round 0.3 · accepted · Accept

In my opinion, this manuscript has been revised with attention to the reviewers' comments and can now be published.

Reviewer 1 ·

Basic reporting

No comment

Experimental design

No comment

Validity of the findings

No comment

Additional comments

I would like to acknowledge the diligence and attention to detail shown by the authors for incorporating the review suggestions into their work. Thank you for your valuable contributions to this topic.